# Adaptation of *Dinoroseobacter shibae* to oxidative stress and the specific role of RirA

**Nicole Beier**[1,2]**, Martin Kucklick**[1,2]**, Stephan Fuchs**[3]**, Ayten Mustafayeva**[1,2]**, Maren Behringer**[1]**, Elisabeth Härtig**[1]**, Dieter Jahn**[1,4]**, Susanne Engelmann**[1,2,4]*

**1** Institute for Microbiology, Technische Universität Braunschweig, Braunschweig, Germany, **2** Microbial Proteomics, Helmholtzzentrum für Infektionsforschung, Braunschweig, Germany, **3** Robert-Koch-Institut, Berlin, Germany, **4** Braunschweig Integrated Centre of Systems Biology (BRICS), Technische Universität Braunschweig, Braunschweig, Germany

* Susanne.Engelmann@helmholtz-hzi.de

**Data Availability Statement:** The mass spectrometry proteomics data were deposited to the ProteomeXchange Consortium

## Abstract

*Dinoroseobacter shibae* living in the photic zone of marine ecosystems is frequently exposed to oxygen that forms highly reactive species. Here, we analysed the adaptation of *D. shibae* to different kinds of oxidative stress using a GeLC-MS/MS approach. *D. shibae* was grown in artificial seawater medium in the dark with succinate as sole carbon source and exposed to hydrogen peroxide, paraquat or diamide. We quantified 2580 *D. shibae* proteins. 75 proteins changed significantly in response to peroxide stress, while 220 and 207 proteins were differently regulated by superoxide stress and thiol stress. As expected, proteins like thioredoxin and peroxiredoxin were among these proteins. In addition, proteins involved in bacteriochlophyll biosynthesis were repressed under disulfide and superoxide stress but not under peroxide stress. In contrast, proteins associated with iron transport accumulated in response to peroxide and superoxide stress. Interestingly, the iron-responsive regulator RirA in *D. shibae* was downregulated by all stressors. A *rirA* deletion mutant showed an improved adaptation to peroxide stress suggesting that RirA dependent proteins are associated with oxidative stress resistance. Altogether, 139 proteins were upregulated in the mutant strain. Among them are proteins associated with protection and repair of DNA and proteins (e. g. ClpB, Hsp20, RecA, and a thioredoxin like protein). Strikingly, most of the proteins involved in iron metabolism such as iron binding proteins and transporters were not part of the upregulated proteins. In fact, *rirA* deficient cells were lacking a peroxide dependent induction of these proteins that may also contribute to a higher cell viability under these conditions.

## Introduction

Aerobic organisms are frequently confronted with reactive oxygen species such as superoxide anion, hydrogen peroxide and the highly reactive hydroxyl radical, which are generated as byproducts of electron transfer reactions such as oxidative phosphorylation. In addition, photosynthetic organisms are exposed to singlet oxygen produced by photophosphorylation under aerobic conditions.

(http://proteomecentral.proteomexchange.org) via the PRIDE partner repository with the dataset identifier PXD013791.

**Funding:** This work was supported by grants of the Deutsche Forschungsgemeinschaft TRR51 to S.E. and D.J. and INST 188/365-1 FUGG to S.E. The funders had no role in study design, data collection and analysis, decision to publish, or preparation of the manuscript.

**Competing interests:** The authors have declared that no competing interests exist.

Reactive oxygen species (ROS) damage membrane fatty acids, proteins and DNA. Both the superoxide radical and hydrogen peroxide are able to oxidise iron sulfur cluster in proteins as shown for dehydratases and regulatory proteins [1, 2]. In this way, enzymes are inactivated and, additionally, ferric iron ($Fe^{3+}$) is released which is rapidly converted to ferrous iron ($Fe^{2+}$). Ferrous iron can reduce $H_2O_2$, which results in hydroxyl radicals in a self-propelling manner. Consequently, an increased iron level can mediate oxidative damage via increased formation of hydroxyl radicals [1]. Adaptation to oxidative stress is characterised by the induction of proteins that are able to help cells to recover from oxidative stress. These proteins are mainly involved in: (i) inactivation of the oxidative stress stimulus, (ii) protection of cellular structures (DNA, proteins, lipids) against oxidative inactivation, and (iii) repair of damaged cellular structures. Protective enzymes to detoxify ROS are superoxide dismutases, catalases, and peroxiredoxins [1, 3]. Moreover, expression of the iron binding protein Dps, thioredoxin, and proteins involved in the repair of Fe-S clusters is induced in response to oxidative stress. Besides, cells employ strategies to protect iron-containing enzymes by replacing $Fe^{2+}$ with $Mn^{2+}$ or by expressing paralogues that do not require iron [1, 4].

The regulatory network mediating the expression of genes involved in oxidative stress response is complex and involves regulators that can directly sense specific ROS. Oxidative modification of these regulators changes their DNA binding activity and thus affects gene expression of target genes. One of the best-studied examples is OxyR, a LysR type transcriptional regulator that senses $H_2O_2$ [5]. It is conserved in Gram-negative bacteria and in a small number of Gram-positive bacteria. LysR-type regulators are characterised by a conserved helix-turn-helix motif involved in DNA binding and a C-terminal sensing and activation domain. OxyR acts as a global regulator and plays a role in the response to peroxide stress and thiol depletion (for review see [6]).

*Dinoroseobacter shibae* is a Gram-negative photoheterotophic bacterium. In contrast to the closely related phototrophic purple bacteria, *D. shibae* performs aerobic anoxygenic photosynthesis. It requires organic substrates such a succinate, glucose and glycerol as carbon and energy sources and uses light driven electron transport as a supplement to respiratory driven electron transport under oxic conditions, which is a very efficient strategy in nutrient restrictive habitats as it exists in marine environments [7, 8]. In the dark, *D. shibae* completely switches to respiratory metabolism using organic substrates such as succinate as electron donors [7]. Recent studies addressed the response of *Rhodobacter sphaeroides* and *Roseobacter denitrificans* to photooxidative stress [9, 10]. Microorganisms that are frequently faced with photooxidative stress have evolved protection mechanisms such as the use of quenchers (e.g. carotenoids) and scavengers (e.g. glutathione). Interestingly, in *R. sphaeroides*, adaptation to singlet oxygen seems to utilise similar pathways as after exposure to high intensities of light but different from adaption to hydrogen peroxide. Hence, distinct regulatory mechanisms exist that are involved in the adaptation to different ROS [10, 11]. RpoE mainly regulates the response to singlet oxygen, while the peroxide stress response is modulated by OxyR [9–11]. In *D. shibae* belonging to the *Roseobacter* group, a change from the heterotrophic growth in the dark to the photoheterotrophic growth in the light has been shown to be accompanied by the induction of proteins typically induced by singlet oxygen [12]. Likewise, homologues of the alternative sigma factors RpoE, RpoH1, and RpoH2 are upregulated under these conditions. Hence, it seems very likely that these alternative sigma factors play a similar role in response to singlet oxygen for *D. shibae* as described for *Roseobacter spec.* and for *R. denitrificans* [9, 10, 12]. However, specific data on the oxidative stress response of *D. shibae* were missing so far. In the present study, we investigated *D. shibae*´s adaptation to different kinds of oxidative stress to identify proteins that are generally induced by oxidative stress and proteins

## Materials and methods

### Bacterial strains and growth conditions

*D. shibae* DFL-12 was cultivated in artificial seawater medium supplemented with 16.9 mM succinate [12] at 30˚C and 160 rpm. For stress experiments, cells were grown to an optical density at 578 nm ($OD_{578}$) of 0.5 and subsequently exposed to the different oxidants (hydrogen peroxide (10–30 mM), paraquat (10–90 μM), and diamide (0.5–1 mM)). Cells were harvested by multiple centrifugation steps before (0 min) and after stress exposure at different time points (30, 60, 120 and 180 min) and disrupted by cell homogenisation (FastPrep-24 ™, MP Biomedicals). To analyse the effect of RirA on peroxide induced gene expression in *D. shibae*, a *rirA* deletion mutant was used (for mutant construction see S1 File). All experiments were performed in triplicates.

### Protein digestion and LC-MS/MS analyses

Aliquots of 20 μg protein crude extracts were separated by one-dimensional SDS polyacrylamide gel electrophoresis [13] using higher concentrations of SDS, ammonium persulfate, and TEMED (see S1 File). In-gel digestion of proteins was carried out as described previously [14]. Each lane was divided into eight subsamples with similar protein amounts which were densitometrically determined using AIDA image analysis software (version 4.15., Raytest Isotopenmeßgeräte GmbH, Straubenhardt, Germany). In-gel digestion of proteins was performed in 50 mM Tris/HCl (pH 7.6) and 1 mM $CaCl_2$. Resulting peptides were extracted and desalted according to Lassek and coworkers [15] starting with an additional acetonitrile extraction step.

For LC-MS/MS analyses, a nanoAQUTY UPLC System (Waters Corporation, Milford, MA, USA) was coupled to an LTQ Orbitrap Velos Pro mass spectrometer (Thermo Fisher Scientific Inc., Waltham, Massachusetts, USA). Peptides from each gel piece were solved in 3% acetonitrile and 0.1% formic acid, ultracentrifuged and loaded onto a BEH C18 column, 130 Å, 1.7 μm, 75 μm x 250 mm at a flow rate of 0.35 μl/min (Waters Corporation, Milford, MA, USA). Elution of peptides from the column was performed using a 222 min gradient and MS scans were conducted in the FT-MS mode (for more details see S1 File).

### MS/MS data analysis and statistics

MS/MS raw files were analysed using MaxQuant (Max Plack Institute of Biochemistry, Martinsried, Germany, www.maxquant.org, version 1.5.2.8) and the following parameters: peptide tolerance: 5 ppm; tolerance for fragment ions: 0.6 Da; variable modification: methionine oxidation, fixed modification: carbamidomethylation; a maximum of three modifications per peptide was allowed; the fixed FDR was set to 1%. All samples were searched against a database containing all protein sequences of *D. shibae DLF 12* extracted from NCBI at 05/09/16 with a decoy mode of reverted sequences and common contaminants supplied by MaxQuant. A label-free quantification mode was selected using MaxQuant LFQ-intensities [16].

A protein was considered when it was detected by at least two unique peptides each with a minimum of two MS/MS scans from at least two MS samples (replicates or samples) of a proteomics project. Statistical analysis was performed by using Perseus software (Version 1.5.0.15, www.maxquant.org).

## Associated data

The mass spectrometry proteomics data were deposited to the ProteomeXchange Consortium (http://proteomecentral.proteomexchange.org) via the PRIDE partner repository [17] with the dataset identifier PXD013791 (https://www.ebi.ac.uk/pride/archive username: reviewer44802@ebi.ac.uk password: nemaRn9C (select "profile" and "Review Submission")).

## Determination of intracellular ATP concentrations

Samples (50 μl) of exponentially growing *D. shibae* cells ($OD_{578}$ = 0.5) cultivated in seawater medium were taken at different time points before and after exposure to 10 mM hydrogen peroxide (20, 30, 60, 120, and 180 min), mixed with 450 μl 90% DMSO, incubated for 2 min at room temperature and subsequently stored at -80˚C. The ATP concentration was determined as described previously [18]. Bioluminescent signals were detected by a luminescent image analyser (LAS-300, Fujifilm) and analysed using the AIDA image analysis software (version 4.15., Raytest Isotopenmeßgeräte GmbH, Straubenhardt, Germany). The amount of ATP in each sample was determined using an ATP standard and referred to $OD_{578}$.

## Results and discussion

### Growth behavior of *D. shibae* in response to alternative sources of oxidative stress

*D. shibae* was cultivated in seawater medium with succinate as sole carbon source in the dark. At exponential growth phase ($OD_{578}$ = 0.5), cells were exposed to different concentrations of hydrogen peroxide, paraquat and diamide. We applied increasing concentrations of hydrogen peroxide ranging from 10 to 30 mM to induce peroxide stress, of diamide from 0.5 to 1 mM to induce thiol stress, and of paraquat from 10 to 90 μM to induce superoxide stress (S1 Fig). Exposure to hydrogen peroxide and diamide resulted in a significant growth rate reduction during the first 30–60 minutes followed by a resumption of growth depending on the applied concentration. In contrast, exposure to the different concentrations of paraquat caused a similar growth inhibition without recovery within the analysed time period (Fig 1A, S1 Fig). A very similar growth behavior in response to these oxidants has been observed for *S. aureus* [19], which was reflected by its protein synthesis pattern [19, 20]. To study the influence of these oxidants on the proteome pattern of *D. shibae*, we applied 10 mM $H_2O_2$, 15 μM paraquat or 0.5 mM diamide that induced a growth rate inhibition by 70, 55, and 24% within the first 30 to 60 minutes during exposure to the three oxidants. This is followed by an increase of growth rates to 80% in the presence of hydrogen peroxide and to 100% in the presence of diamide (each compared to unstressed cells) (Fig 1A).

### Peroxide, superoxide and thiol stress resulted in global changes in the protein expression pattern of *D. shibae* mainly attributed to protein and DNA damage, energy limitation and disorder of iron homeostasis

To define and compare the peroxide, superoxide and thiol stress stimulon in *D. shibae*, cytoplasmic proteins were prepared from cells before and at different time points after exposure to the respective oxidants (30, 60, 120, and 180 min) and further analysed by GeLC-MS. In this way, we were able to identify overall 2580 *D. shibae* proteins. Adaptation to the three stimuli was characterized by dramatic changes in the protein expression pattern within the first hour after stress exposure. Altogether 444 protein profiles showed statistically significant differences in amount by at least 1.5 fold (log2 fold changes of ≥ 0.58 for induced proteins and of ≤ -0.58 for repressed proteins) in response to at least one stimulus using one way ANOVA. 195 of

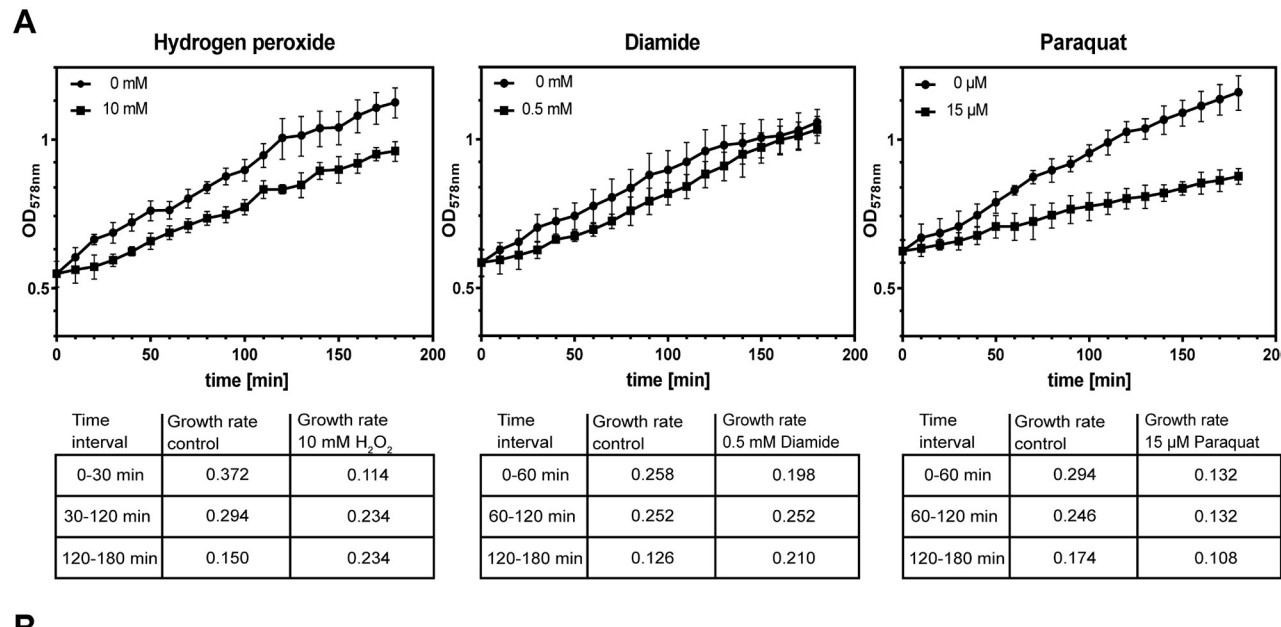

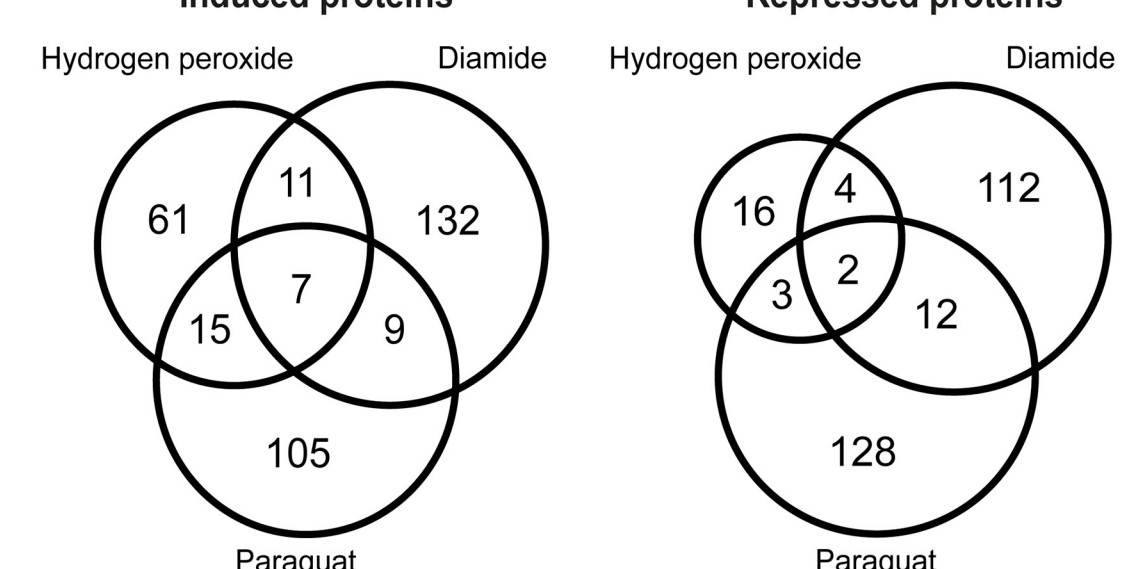

**Fig 1. Adaptation of *D. shibae* to different kinds of oxidative stress. (A)** Effect of 10 mM $H_2O_2$, 0.5 mM diamide and 15 μM paraquat on growth of *D. shibae*. Cells were grown aerobically in artificial seawater medium. At an OD at 578 nm ($OD_{578}$) of 0.5 (exponential growth phase), the different oxidants were added to the cultures. For each stressor, experiments were performed in triplicates and growth curves displayed represent mean values (± SD). **(B)** Overlap of marker proteins significantly induced or repressed by the different stimuli.

these proteins were upregulated during stress (S1 Table). Additionally, 153 proteins were missing in unstressed cells and were only detectable in cells exposed to at least one stimulus. This resulted in 159 protein profiles showing an induction during diamide stress followed by paraquat (n = 136) and hydrogen peroxide (n = 94). In parallel, the amount of 130, 145, and 25 proteins was negatively affected by at least 1.5 fold in response to diamide, paraquat and hydrogen peroxide, respectively (S1 and S2 Tables).

A comparison of the proteome signatures obtained under the different conditions revealed proteins generally induced or repressed by oxidative stress and proteins that were rather

specifically induced or repressed by only one stimulus. The amount of 63 proteins altered in response to two or three of the oxidants and were considered as general ROS responsive proteins while 532 differently expressed proteins were determined as specific for the respective ROS applied in the present study (Fig 1B). The overlap among the three stimulons was very small. One might suspect that the applied oxidants act on completely different ways in *D. shibae*. This was expected as these ROS are chemically diverse and as such are quenched utilising diverse pathways. Consequently, only seven proteins showed increased expression profiles in response to all three stimuli (Fig 1B). These were the ferritin like protein Dshi_0107, the iron-regulated protein Dshi_0563, the small heat shock protein Hsp20 (Dshi_2892), three enzymes belonging to the 2-Keto-3-desoxy-6-phosphogluconate (KDPG) pathway (Dshi_1768, Dshi_1769, Dshi_1684) and one hypothetical protein (Dshi_2778) (S2 Table). In contrast, amounts of the processing peptidase Dshi_1147 and the 5-aminolevulinic acid synthase Dshi_1182 were decreased in response to the three stimuli (Fig 1B, S2 Table). Reduced activity of Dshi_1182, which catalyses an early step of porphyrin synthesis, will have consequences for the activity of down-stream synthesis pathways such as cobalamin, heme, and bacteriochlorophyll biosynthesis. Eleven additional proteins was induced by peroxide and thiol stress. Fifteen induced proteins were shared by hydrogen peroxide and paraquat stress and nine proteins by paraquat and diamide stress (Fig 1B).

While the number of globally affected proteins was small, results indicated that restrictions in iron and energy metabolism, DNA and protein damage and reduced tetrapyrrole biosynthesis may be common effects in *D. shibae* after exposure to oxidative stress. When considering the functional protein categories, similarities in different types of stress response of *D. shibae* to the different oxidants became even more apparent. For these analyses, we selected proteins with significantly changed expression profiles (at least 1.5 fold) in response to at least one stimulus. All proteins that were only detectable in response to at least one stimulus but missing in unstressed cells were added to this list (S1 Table). The most apparent functional protein clusters were related to protein and DNA-repair, iron metabolism, energy- and carbon metabolism, electron transport chain, porphyrin biosynthesis and related pathways, stress proteins and translation. Altogether, 24 proteins were allocated to protein repair and degradation, 36 to carbon and energy metabolism, 16 to iron metabolism, 22 to the electron transport chain, 21 to DNA replication, recombination and repair and 18 to porphyrin biosynthesis including cobalamin, heme and bacteriochlorophyll biosynthesis, 58 to transport processes and 14 proteins are so called stress proteins. Expression profiles of these proteins in response to the different oxidants were compared (Figs 2–4, S1 Table).

**(i) Iron metabolism.**   Six out of 16 proteins involved in iron metabolism showed induction profiles by at least 1.5 fold in response to all stimuli but did not always meet the criteria for statistical significance (Fig 3, S1 Table). Among these proteins are mainly iron scavenging proteins as for instance the heme binding proteins HmuS, a ferritin like protein (Dshi_0107) and the substrate binding component of an $Fe^{3+}$ ABC transporter (Dshi_2021). Proteins involved in Fe-S cluster assembly (Dshi_1069, Dshi_1730) were mainly induced by diamide whereas the presence of hydrogen peroxide and paraquat had no or repressing effects on the amount of these proteins (S1 Table). Diamide as a thiol oxidizing agent destroys specifically iron sulfur cluster within proteins which may also occur in response to the concentrations of $H_2O_2$ applied here but not to paraquat. Notably, three potential iron regulators with high similarities to Irr (Dshi_1011), RirA (Dshi_1660) and IscR (Dshi_1633) were also among the differentially regulated proteins. While the amount of RirA and Irr was generally reduced by oxidative stress, IscR accumulated in *D. shibae* after treatment with paraquat and diamide but not with peroxide (Fig 5). The increased expression of iron binding systems in *D. shibae* in response to oxidative stress might diminish the intracellular unincorporated iron to attenuate

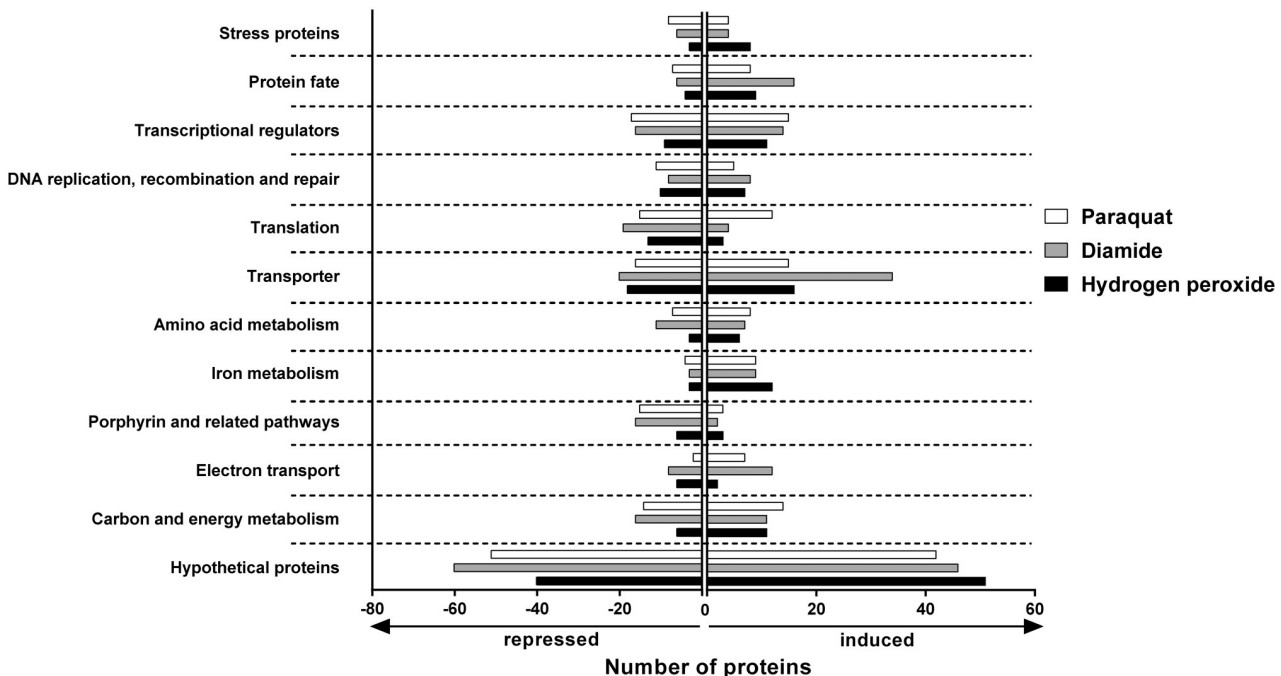

**Fig 2. Selected functional categories of proteins with significantly different amounts in response to the various oxidants.** Bars represent the number of proteins within the given category whose amount significantly changed in response to the respective stimulus by at least 1.5 fold.

the Fenton reaction and its toxic effects in the cells. This function has been particularly described for Dps (DNA binding protein from starved cells) which belongs to the ferritin protein family [21]. In *Escherichia coli*, *dps* expression is regulated by OxyR and RpoS and is thus strongly upregulated in response to oxidative stress and starvation (for review see [22]). Interestingly, *D. shibae* possesses three Dps encoding genes (Dshi_3839, Dshi_3912, Dshi_4223) which are all localized on plasmids. Solely the amount of Dshi_3912 strongly accumulated under these conditions and could take on this role. It still remains open whether iron scavenging proteins such as ferritin and heme binding proteins (Dshi_0107, Dshi_0086, Dshi_0573, Dshi_0574) also play a role in this process in *D. shibae* (Fig 3, S1 Table).

**(ii) Protein repair and degradation.** Protein damage mainly induced by oxidation of specific amino acids is known to affect cell viability during oxidative stress in living cells. As expected, a relatively high number of proteins (n = 21) involved in protein modification, repair and degradation is within the group of inducible proteins (S1 Table). The majority was upregulated in response to thiol stress (n = 16) followed by peroxide (n = 9) and super oxide stress (n = 8) (S2 Table, Fig 4). Interestingly, only three proteins have been found to be upregulated by all three stimuli simultaneously: the small heat shock proteins Hsp20 and Hsp33 and an additional IbpAB like protein (S2 Table). These proteins act as chaperones and are involved in the protection of denatured proteins against irreversible aggregation [23, 24]. In addition, the chaperone and ATPase ClpB accumulated during exposure with diamide and peroxide. And finally, various proteases and peptidases (n = 11) were affected, however, predominantly in response to thiol stress. Three of them (Dshi_3870, Dshi_1540 and Dshi_1773) were even specifically induced by thiol stress (Fig 4, S1 Table). This suggests that the ability to cope with protein damage might be of greater significance on *D. shibae* survival under thiol and peroxide stress than under superoxide stress.

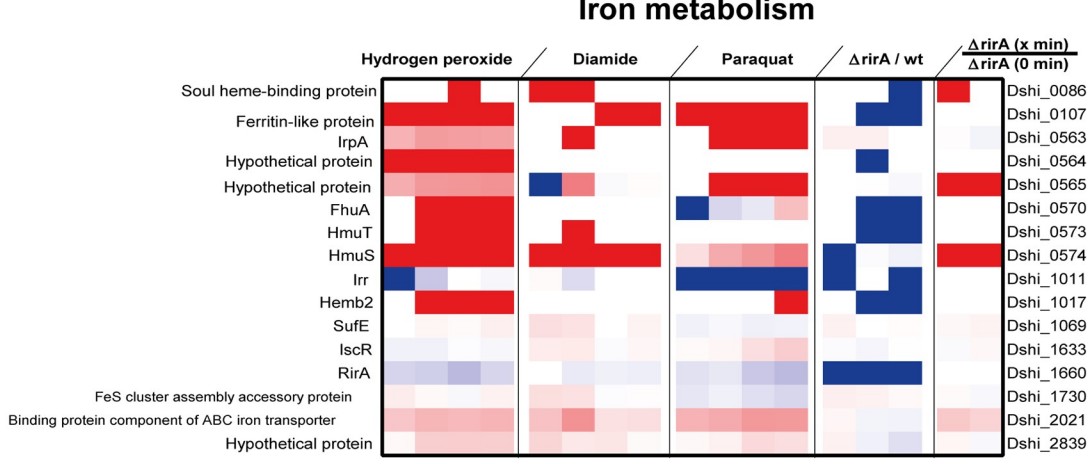

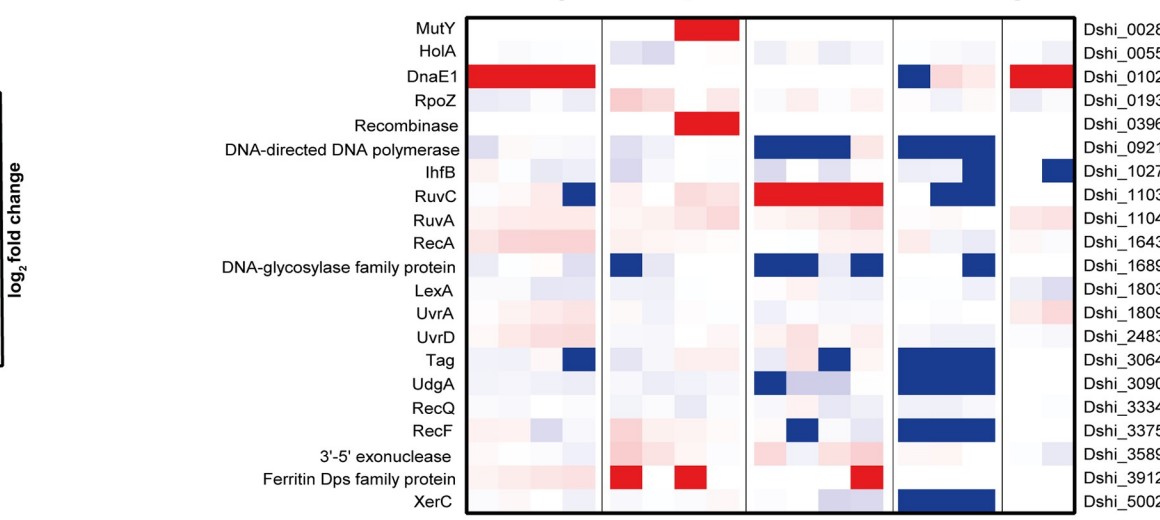

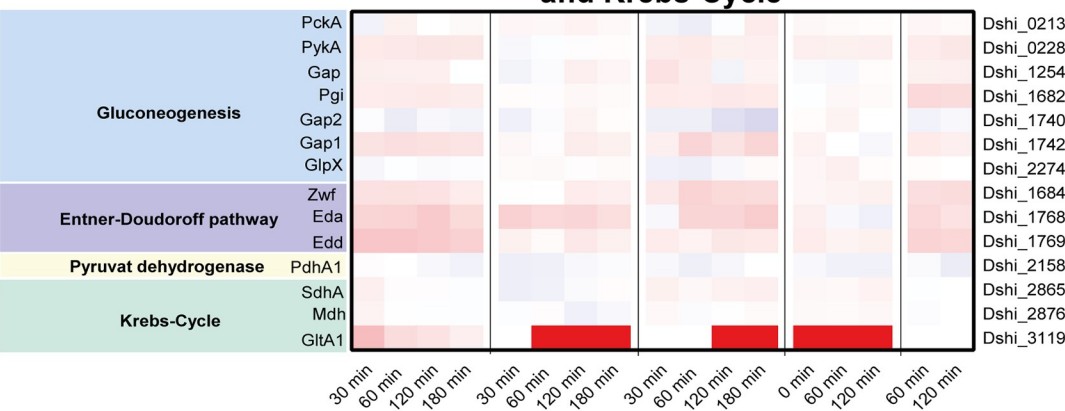

**Fig 3. Differently expressed proteins in response to oxidative stress associated with iron metabolism, DNA-replication, recombination and repair and carbon and energy metabolism.** Heat maps showing the expression profiles of significantly differently expressed proteins in response to the various oxidants (blue—downregulated, red–upregulated). Only proteins with at least 1.5 fold change in expression to at least one of the stimuli (10 mM hydrogen peroxide, 0.5 mM diamide, and 15 µM paraquat) were depicted. For the *rirA* mutant, expression profiles for these proteins in response to peroxide stress (10 mM H$_2$O$_2$) are shown. In addition, the amount of the respective proteins in the *rirA* mutant was compared to that in the wild type. Each experiment was performed in triplicates.

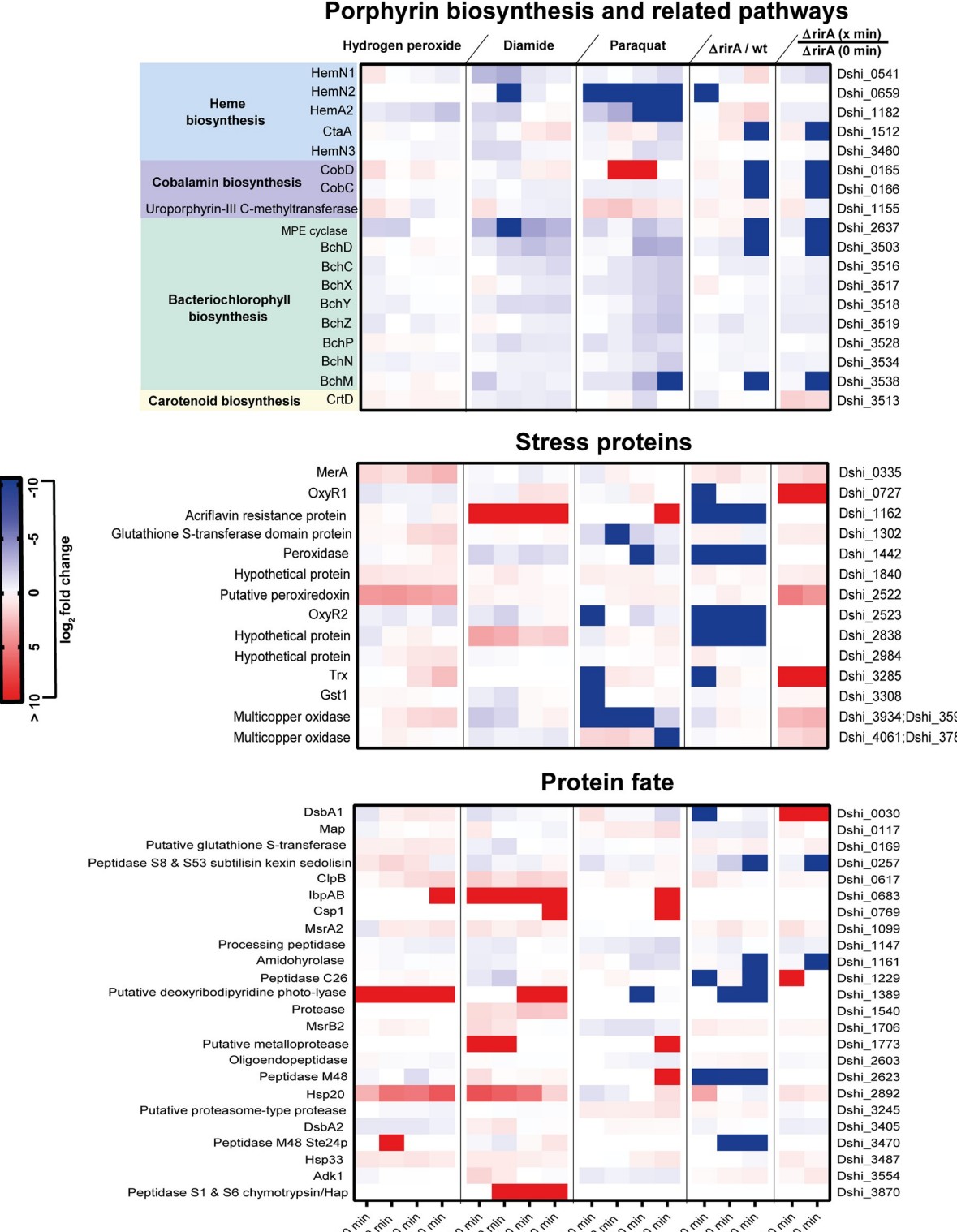

**Fig 4. Differently expressed proteins in response to oxidative stress belonging to the functional categories porphyrin biosynthesis, protein fate and stress proteins.** Heat maps showing the expression profiles of significantly differently expressed proteins in response to the various oxidants (blue—downregulated, red–upregulated). Solely proteins are presented the amount of which was statistically significantly changed by at least 1.5 fold in response to at least one of the stimuli applied in this study (10 mM hydrogen peroxide, 0.5 mM diamide, and 15 μM paraquat). For the *rirA* mutant, expression profiles for these proteins in response to peroxide stress (10 mM $H_2O_2$) are shown. In addition, the amount of the respective proteins in the *rirA* mutant was compared to that in the wild type. Each experiment was performed in triplicates.

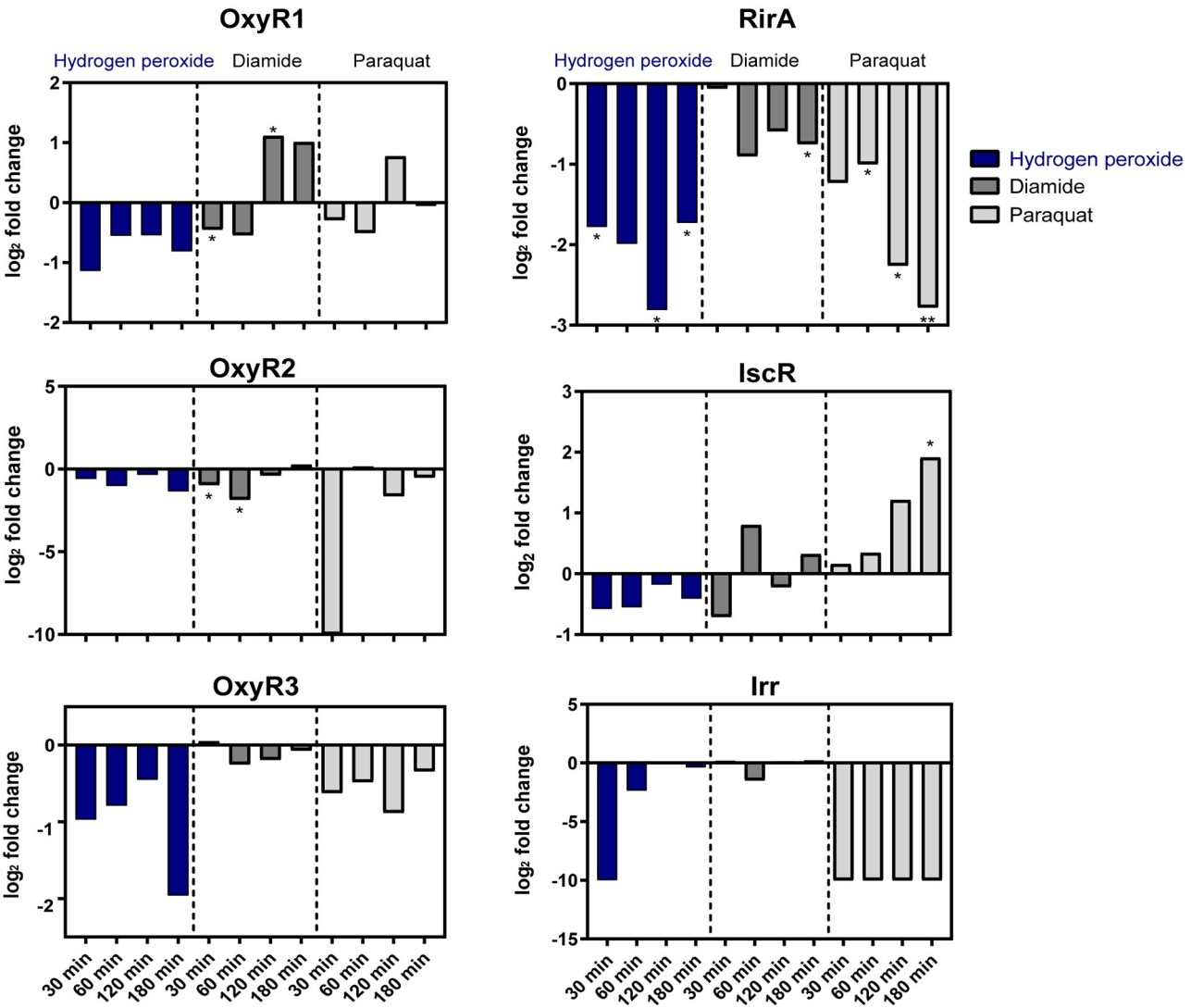

**Fig 5. Expression profiles of the iron-responsive regulators RirA, IscR and Irr and the OxyR homologues OxyR1, OxyR2 and OxyR3 of *D. shibae* in response to the different oxidants.** Cytoplasmic protein extracts of *D. shibae* wild type were prepared from cells before and after exposure to 10 mM hydrogen peroxide, 0.5 mM diamide and 15 μM paraquat and analyzed by a GeLC-MS/MS approach. The histograms show LFQ intensities as log2 fold changes of the amount of the three *D. shibae* OxyR homolgues OxyR1 (Dshi_0727), OxyR2 (Dshi_2523), and OxyR3 (Dshi_3802) and of the potential iron-responsive regulators RirA (Dshi_1660), IscR (Dshi_1633), and Irr (Dshi_1011). Each experiment was performed in triplicates.

**(iii) DNA repair and modification.** DNA damage was suggested to be a general problem *D. shibae* encounters under oxidative stress conditions. This is strongly suggested by the fact that the amount of the SOS regulator LexA was diminished by all three oxidants. In line with this, twelve proteins showing an induction profile under at least one of these conditions are involved in DNA repair and modification. RuvA and RuvC were more generally induced during oxidative stress while the amount of Uvr was specifically affected by peroxide stress. The dsDNA heteroduplex molecule, which is generated by pairing of ssDNA with its undamaged homologous regions, represents the substrate of the Uvr repair system and is triggered by the activity of RecA [25]. The latter showed an induced expression profile in response to peroxide stress but also in response to superoxide. Notably, other components of the Rec system, the

major repair system for single strand DNA gaps and double strand breaks [26], remained mainly unaffected under these conditions in *D. shibae* (Fig 3, S1 Table).

**(iv) Carbon and energy metabolism.** Proteins belonging to carbon and energy metabolism such as glyceraldehyde dehydrogenase, aconitase, succinate dehydrogenase and α-ketoglutarate dehydrogenase are known targets of oxidative damage and consequently inactivated under these conditions [18, 27, 28]. In this way the energy metabolism is disturbed which is accompanied by a significant drop in ATP production [18]. In superoxide stressed *Escherichia coli* cells, an increased ratio of NADPH:NADH and an accumulation of α-ketoglutarate has been observed [28]. In peroxide treated *D. shibae* cells, the ATP concentration dropped to 60% within 60 min compared to cells immediately before exposure to peroxide stress (Fig 6). Interestingly, a similar decline of the ATP level has been observed in *D. shibae* cells lacking oxygen as terminal electron acceptor [29]. In the untreated control culture, the cellular ATP-concentration slightly increased within 120 min (1.2 fold) and decreased afterwards which is closely related to the transition into the stationary growth phase (Fig 6). The overall induction of proteins belonging to the KDPG pathway, which was striking for peroxide and superoxide stressed cells, might be a direct reaction to this limitation. At the same time key enzymes of gluconeogenesis and pentose phosphate pathway were repressed suggesting that energy consuming processes are decreased (Fig 3, S1 Table). Glyceraldehyde dehydrogenase, which represents a key enzyme of the KDPG pathway and gluconeogenesis, has been shown to be a subject of irreversible oxidation induced by oxidative stress in prokaryotes [18]. The genome of *D. shibae* encodes three glyceraldehyde dehydrogenase isoenzymes (Dshi_1254, Dshi_1740, and Dshi_1742) all of which were differently expressed in response to oxidative stress. While two of these enzymes (Dshi_1254, Dshi_1742) accumulated under these conditions, Dshi_1740 was clearly repressed suggesting that the first two might have rather catabolic function. In addition, paraquat and diamide reduced enzymes involved in *de novo* synthesis of amino acids

**Fig 6. Effects of peroxide stress on the intracellular ATP level in *D. shibae*.** *D. shibae* wild type was grown in artificial seawater medium to an OD$_{578}$ of 0.5 and exposed to 10 mM hydrogen peroxide. Cellular ATP concentration was determined in peroxide stressed cells as well as in untreated cells at different time points.

(S1 Table) causing an overall restriction of amino acids in stressed cells. This in turn should provoke an increase in uncharged tRNAs, a signal which triggers the stringent response. This is a fundamental global adaptation strategy universally distributed in eubacteria. The major function of this response is to adjust the cell's biosynthetic machinery to the availability of energy and required precursors [30] and is mainly characterized by downregulation of the translational machinery, which was also striking, for thiol and superoxide stressed cells (S1 Table).

**(v) Porphyrin biosynthesis and related pathways.** The general repression of Dshi_1182 providing 5-aminolevulinic acid as a precursor for heme as well as cobalamin and bacterio-chlorophyll biosynthesis provided first hints that the production of these molecules is restricted under these conditions (S1 Table). Interestingly, the amount of altogether 17 proteins linked to heme (n = 5), cobalamin (n = 3) and bacteriochlorophyll (n = 9) biosynthesis in *D. shibae* were identified to be significantly suppressed by thiol and superoxide stress, thus being the more efficient effectors by repressing 16 and 15 proteins (Fig 4, S1 Table). In response to peroxide stress, only seven of them have been found with lower quantities. Interestingly, a reduced transcriptional activity of photosynthetic and heme biosynthesis genes was also observed immediately in response to light exposure which is accompanied by photooxidative stress induced by singlet oxygen. The transcription factor PpsR is suspected to be responsible for this phenomenon [12] and possibly measures the presence of oxygen via heme. The particularly strong effect of diamide, which is a thiol oxidizing agent, on *D. shibae* cells grown in the dark provides further evidence for a thiol dependent regulatory mechanism that might play a role in regulation of these genes dependent on oxygen tension and completely independent of light. This phenomenon is widely distributed among aerobic anoxygenic phototrophs [31–33]. When grown under low oxygen tension, this enables those bacteria to reactivate their photosynthetic activity also in the dark. Interestingly, the sensor kinase RegB that senses redox changes through multiple mechanisms was among the proteins that accumulated after exposure to diamide (S1 Table). The cytosolic part of the protein contains a redox sensitive cysteine that can be oxidised to disulfide bond or to sulfenic acid, thereby inhibiting RegB autophosphorylation [34, 35]. In *Rhodobacter capsulatus* and *R. sphaeroides* the transcriptional regulator RegA (PrrA) has been shown to be involved in the regulation of photosynthetic genes [36]. It is reasonable to assume that in *D. shibae* the system also directly affects the expression of these genes. The rather small effects observed for the expression of photosynthetic genes in response to 10 mM $H_2O_2$, also known as an inducer of thiol oxidations, may indicate a significantly lower rate of reversible thiol oxidations under these conditions.

**(vi) Oxidative stress proteins.** Finally, proteins known to be essential in stress adaptation were also among the proteins with changing amounts in response to the oxidants. These are for instance peroxiredoxins, glutathione-S-transferases, a thioredoxin-like protein and a peroxidase like protein (Fig 4, S1 Table). Interestingly, proteins such as catalase, superoxide dismutase, thioredoxin, glutathione synthetase and glutathione reductase known to be directly involved in degradation of oxidants or in repair of damaged cell structures remained almost unaffected under these conditions. This has also been shown for other bacteria where in particular detoxifying enzymes such as catalase and superoxide dismutase were expressed in constitutively high amounts in the absence of oxidants [20, 37]. Similar expression kinetics of these proteins in *D. shibae* may provide an explanation for the observed high resistance of the bacterium to oxidative stress. OxyR is known as a major regulator for genes encoding oxidative stress proteins in Gram-negative bacteria. The genome of *D. shibae* codes for three OxyR like proteins Dshi_0727 (OxyR1), Dshi_2523 (OxyR2) and Dshi_3802 (OxyR3) with 39%, 38%, and 37% identity to OxyR in *E. coli*. Dshi_3802 possesses two cysteine residues that might correspond to $Cys_{199}$ and $Cys_{208}$ of *E. coli* OxyR. Both residues are involved in disulfide bond

formation in *E. coli* OxyR and essential for its activity [38–43]. By contrast, the two other OxyR homologues contain only one of these cysteine residues: in Dshi_2523 this residue roughly corresponds to $Cys_{199}$ in *E. coli* while in Dshi_0727 $Cys_{208}$ seems to be conserved. Expression profiles of the three OxyR proteins in response to peroxide, disulfide and superoxide stress showed rather reduced protein levels. Only OxyR1 significantly accumulated under thiol stress conditions (Fig 5, S1 Table). However, this does not imply an effect on the activity of these regulators.

### Specific role of the iron-responsive regulator RirA in adaptation of *D. shibae* to peroxide stress

In rhizobia, RirA is a Fe-S protein with a transcriptional repressor function for many iron-regulated genes to maintain iron homeostasis [44, 45]. The RirA protein in *D. shibae* possesses 44% amino acid sequence identity to RirA in *Rhizobium leguminosarum*. Its amount was significantly reduced by at least two-fold in response to the three oxidants (Figs 3 and 5). In addition, induction of proteins involved in iron metabolism was the most striking response to peroxide stress in *D. shibae*. Hence, RirA may be a good candidate in *D. shibae* to adjust the global gene expression program to the requirements during oxidative stress. We tested the growth of a *rirA* deficient strain in *D. shibae* under control conditions and during exposure to hydrogen peroxide, diamide and paraquat. Accordingly, RirA deficiency did not lead to an altered growth phenotype under control conditions (Fig 7). However, immediately after imposition of peroxide stress, the *rirA* mutant was significantly more viable than the wild type: while the wild type completely stopped growth within the first 30 minutes during exposure to hydrogen peroxide, the growth rate of the *rirA* mutant was only reduced by 50% (Fig 7, Table 1). Interestingly, this effect was not observed under thiol and super oxide stress. Introduction of *rirA* localized on a plasmid in the deletion mutant restored the wild type phenotype in response to peroxide stress (Fig 7, Table 1).

To identify proteins that might contribute to the observed growth behavior related to *rirA* deletion, the global impact of RirA on protein expression in *D. shibae* was studied before and 60 and 120 min after exposure to hydrogen peroxide. The amount of 163 proteins was found to be significantly changed in the absence of RirA. In addition, 832 proteins were missing either in the wild type or in the *rirA* mutant at least one time point. Only 139 proteins were

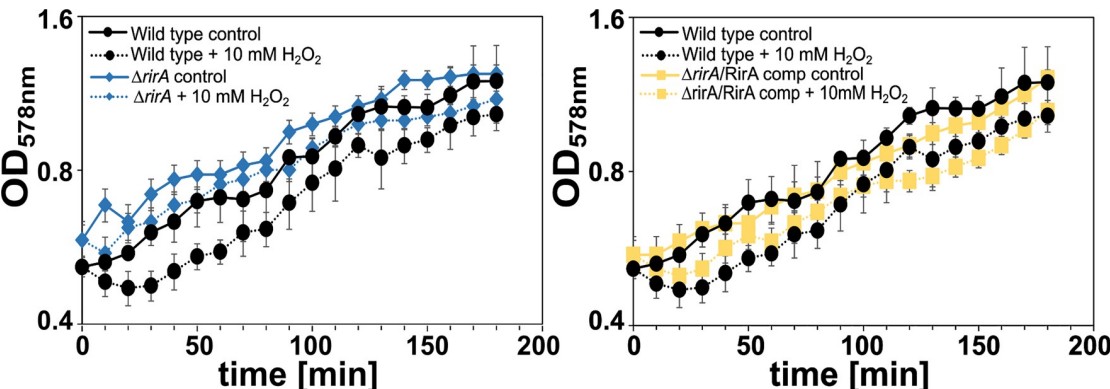

**Fig 7. Growth of *D. shibae* wild type, Δ*rirA* and Δ*rirA*/RirA complemented strain grown under control and peroxide stress conditions.** Cells were grown under aerobic conditions in artificial seawater medium in the dark. At an OD at 578 nm ($OD_{578}$) of 0.5 (exponential growth phase), 10 mM $H_2O_2$ (peroxide stress) were added to the cultures. Growth was monitored for the different strains (wild type, Δ*rirA* and Δ*rirA*/RirA complemented strain) under control conditions and in response to peroxide stress. Each experiment has been performed in triplicates and growth curves based on mean values (±SD) are shown.

**Table 1. Growth rates of wild type, Δ*rirA* and Δ*rirA*/RirA complemented strain grown under control and peroxide stress conditions[a].**

| | wild type | | | | Δ*rirA* mutant | | | | Δ*rirA*/RirA complemented strain | | | |
|---|---|---|---|---|---|---|---|---|---|---|---|---|
| | 0 mM $H_2O_2$ | | 10 mM $H_2O_2$ | | 0 mM $H_2O_2$ | | 10 mM $H_2O_2$ | | 0 mM $H_2O_2$ | | 10 mM $H_2O_2$ | |
| time interval | μ[c] | SD[d] | μ[c] | SD[d] | μ[c] | SD[d] | μ[c] | SD[d] | μ[c] | SD[d] | μ[c] | SD[d] |
| 0–30 min | 0.30 | 0.01 | -0.34 | 0.14 | 0.33 | 0.13 | 0.16 | 0.05 | 0.25 | 0.14 | -0.19 | 0.11 |
| 30–120 min | 0.34 | 0.09 | 0.41 | 0.08 | 0.27 | 0.03 | 0.28 | 0.04 | 0.27 | 0.03 | 0.27 | 0.01 |

[a]Strains were grown in artificial seawater medium to an $OD_{578}$ of 0.5 and then exposed to 10 mM $H_2O_2$. Growth of the control and stressed cultures was monitored by measuring the optical density at 578 nm and growth rates were determined for the indicated time intervals. The experiments have been performed in triplicates.

[b]time interval after exposure to stress.

[c]growth rates (μ) are given in $h^{-1}$.

[d]standard deviation.

found to be upregulated in the *rirA* mutant: 79 proteins under control conditions and 60 solely in response to peroxide stress. Candidates that might mediate the observed higher resistance of the *rirA* mutant under peroxide stress conditions were expected to be among these proteins. The bacterial DNA recombination protein RecA, ClpB, the thioredoxine like protein Dshi_2984, Hsp20, a putative glutathione S-transferase (Dshi_0169) and peptidase M16 like protein Dshi_0362 (S3 Table) were of particular interest as they may contribute to protection, repair and degradation of proteins and DNA targeted by oxidative stress. While thioredoxin-like proteins and glutathione S-transferases are proteins with a putative role in repair of oxidized cysteines in proteins, ClpB and Hsp20 are canonical chaperons and thus involved in refolding and removal of aggregated proteins. It has been demonstrated that deficiency of these proteins resulted in growth defects under oxidative stress in several bacteria (for reviews see [46, 47]). In addition, several hypothetical proteins were detected among these proteins. Their precise role in oxidative stress resistance would be interesting to study (S3 Table).

Notably, with the exception of the iron binding protein IscA (Dshi 2349) and an MntR homologue (Dshi 3611), none of the identified proteins assigned to the iron metabolism were among the proteins derepressed by a loss of RirA (Fig 3, S3 Table). Detailed transcriptional analyses of *hemB2* already suggested that only at low iron conditions, the absence of RirA led to derepression of *hemB2* transcription. Under high iron conditions, IscR may be the major repressor of *hemB2* transcription [48]. Data presented here provide first evidence that this might be the case not only for HemB2 but also for other proteins associated with the iron metabolism (e. g. SOUL heme-binding protein Dshi_0086, ABC-type cobalamin/Fe3+-sidero-phores transport protein FhuA, hemin import ATP-binding protein HmuV and hemin-binding periplasmic protein HmuT) and has to be analysed in more detail. Surprisingly, peroxide induced expression of some of these proteins was dependent on RirA: Of the twelve peroxide induced proteins involved in iron metabolism (S1 Table), six proteins (SOUL heme-binding protein, ABC-type cobalamin/Fe3+-siderophores transport protein FhuA, protein DUF1111 protein, TonB-dependent heme/hemoglobin receptor HemB2, hemin-binding periplasmic protein HmuT, ferritin-like protein Dshi_0107) were suppressed in *rirA* mutant cells (S3 Table). Hence, it should also be considered that elevated levels of these proteins in peroxide stressed wild type cells may disturb iron homeostasis under high iron conditions. Consequently, Fenton reaction and its toxic effects may impair viability of these cells.

## Conclusions

Exposure to either peroxide, superoxide or thiol stress resulted in global changes in protein expression pattern in *D. shibae* mainly related to protein and DNA damage, energy limitation

and disorder of iron homeostasis. Notably, the amount of tetrapyrrole biosynthesis enzymes including those involved in bacteriochlorophyll and heme biosynthesis was reduced by each stimulus, albeit with different intensities. This might be mediated by a thiol switch regulatory mechanism enabling *D. shibae* to express photosynthetic genes dependently on oxygen tension and completely independently of light. It seems very likely that the two-component system RegAB is essential for this process. Further characterization of the RegAB regulon is of special interest.

The response of *D. shibae* to hydrogen peroxide was less pronounced compared to paraquat and diamide, which became apparent by a significantly lower number of induced and repressed proteins in peroxide stressed cells. This might be due to the fact that the applied concentration of hydrogen peroxide induced a stronger growth rate inhibition. While in the presence of 10 mM hydrogen peroxide the growth rate was reduced by 70%, the applied concentrations of diamide and paraquat caused a reduction by 25 and 50% (Fig 1A). In addition, survival of peroxide treated cells dropped by nearly 50% compared to control cells, which has not been observed for diamide and paraquat. It is thus possible that in hydrogen peroxide treated cells, proteins, lipids, and nucleic acids are more frequently damaged. This might also explain the observed lower expression of biosynthetic genes in response to peroxide stress.

The regulatory network responsible for adjustment of gene expression in *D. shibae* in response to oxidative stress was less elucidated. Interestingly neither RpoE (Dshi_3423) nor RpoH1 (Dshi_2978) and RpoH2 (Dshi_2609) that play an essential role in adaptation to photooxidative stress were significantly induced in response to the different oxidants applied here. The expression pattern of RirA in response to different oxidants and the observed phenotypic changes of the *rirA* mutant support the notion that *D. shibae* RirA is a global transcriptional regulator that possibly senses oxidative stress via an iron sulfur cluster that has recently been shown to be oxygen labile [48]. It represses or activates the expression of genes involved in multiple cellular processes such as protein and DNA repair, energy metabolism and several transport processes and may thus confer adaptive resistance to oxidative stress. However, RirA is among others also responsible for the peroxide responsive gene expression of iron-regulated genes, which could impair the viability of *D. shibae* wild type cells. This effect may be only relevant if iron is present in excessive concentrations, which is rather unusual in the marine environment, the natural habitat of *D. shibae*.

There is evidence that at least 35 genes that are induced during starvation in a light dark cycle might be involved in oxidative stress resistance in *D. shibae* cells [49]. Interestingly, solely three of these proteins were specifically affected by the oxidants used in the present approach suggesting that the remaining proteins might rather be involved in a more general, starvation induced response providing non-growing or stressed *D. shibae* cells with a multiple stress resistance. This phenomenon has also been observed in other bacteria and is regulated by alternative sigma factors such as RpoS or $\sigma^B$ [50–53].

## Supporting information

**S1 Table. Proteins with changing amounts that reached statistical significance in response to at least one stimulus at least one time point.**
(XLSX)

**S2 Table. Overlap of significantly changed proteins in response to hydrogen peroxide, diamide and paraquat at least one time point.**
(XLSX)

**S3 Table. Proteins with changing amounts in the Δ*rirA* mutant that reached statistical significance.**
(XLSX)

**S1 Fig. Effect of oxidative stress on growth of *D. shibae*.** Cells were grown aerobically in seawater medium. At an OD at 578 nm ($OD_{578}$) of 0.5 (exponential growth phase), different concentrations of the oxidants were added to the cultures. $H_2O_2$: 0mM (•), 10 mM (■), 20 mM (▲), and 30 mM (▼). Diamide: 0 mM (•), 0.5 mM (■), 0.65 mM (▲), 0.8 mM (▼), and 1.0 mM (○) Paraquat: 0 μM (•), 10 μM (■), 15 μM (▲), 30 μM (▼), and 90 μM (○). For each concentration one representative growth curve is represented.
(PDF)

**S1 File. Supporting material and methods.**
(PDF)

# Acknowledgments

We are grateful to Daniel Meston and Cornelius Engelmann for critical reading the manuscript. We thank B. Jung for technical assistance.

# Author Contributions

**Conceptualization:** Stephan Fuchs, Susanne Engelmann.

**Data curation:** Nicole Beier, Martin Kucklick, Ayten Mustafayeva, Maren Behringer.

**Formal analysis:** Nicole Beier.

**Investigation:** Nicole Beier.

**Methodology:** Martin Kucklick, Stephan Fuchs.

**Project administration:** Susanne Engelmann.

**Resources:** Susanne Engelmann.

**Supervision:** Stephan Fuchs, Elisabeth Härtig, Dieter Jahn, Susanne Engelmann.

**Writing – original draft:** Susanne Engelmann.

**Writing – review & editing:** Nicole Beier, Martin Kucklick, Stephan Fuchs, Ayten Mustafayeva, Elisabeth Härtig, Dieter Jahn, Susanne Engelmann.

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
