## [Decision Letter · Decision Letter 0]

10 Nov 2020

PONE-D-20-31011

Adaptation of *Dinoroseobacter shibae* to oxidative stress and the specific role of RirA

PLOS ONE

Dear Dr. Engelmann,

As you can see from their comments below, both reviewers felt that your manuscript describes an interesting, and perhaps not surprising, link between RirA and the resistance of  *Dinoroseobacter shibae* to oxidative stress. But both raised serious concerns about the confusing manner in which the experimental data are being presented. Based on these assessments, I am going to ask that you submit a revised version of  the manuscript that adequately and appropriately addresses all of  the comments raised by both of  these reviewers.

We look forward to receiving your revised manuscript!

Sincerely,

R. Martin Roop II, Ph.D.

Academic Editor

PLOS ONE

Journal Requirements:

2.Thank you for stating the following financial disclosure:

 [The funders had no role in study design, data collection and analysis, decision to publish, or preparation of the manuscript.].

Reviewers' comments:

Reviewer's Responses to Questions

**Comments to the Author**

1. Is the manuscript technically sound, and do the data support the conclusions?

Reviewer #1: Partly

Reviewer #2: Partly

2. Has the statistical analysis been performed appropriately and rigorously? 

Reviewer #1: I Don't Know

Reviewer #2: Yes

3. Have the authors made all data underlying the findings in their manuscript fully available?

Reviewer #1: Yes

Reviewer #2: Yes

4. Is the manuscript presented in an intelligible fashion and written in standard English?

Reviewer #1: No

Reviewer #2: Yes

5. Review Comments to the Author

Reviewer #1: This manuscript presents a highly detailed analysis of proteomic changes in the marine bacterium Dinoroseobacter shibae in response to oxidative stress. The authors analyze temporal changes in the bacterial proteome in response to three different types of oxidative stress. In addition, they also examine a mutant lacking the iron-responsive transcriptional repressor RirA. An impressive amount of work has gone into compiling the results presented.

However, the manuscript is rather confusing as written and could benefit from changes in how the data is presented. As stated in the abstract, the authors seek to demonstrate that “RirA dependent proteins are important for oxidative stress resistance”. However, the link of the RirA regulator to the changes in response to oxidative stress (peroxide stress in particular) is lost in the manuscript writing and should be better highlighted.

My major comments are listed below:

1. The authors introduce RirA as a likely iron-responsive regulator (lines 263-264 and 387-388) and seek to address its role in oxidative resistance in D. shibae. Furthermore, in lines 443-444, the authors state that “The most striking response of D. shibae to oxidative stress was the induction of proteins involved in iron metabolism.” It is therefore puzzling to me that the authors choose to highlight not iron-related proteins, but a putative glutathione S-transferase, ClpB and Hsp20 as particularly interesting candidates identified with the mutant in the abstract (lines 40-45). It is also not made clear why they are particularly interesting; I think it would be a stronger paper if the focus in the abstract remains in the context of iron- related RirA mechanisms.

Line 40: Adding the term “iron-responsive” in front of the term “regulator RirA” may be more informative for the reader.

2. Fig. 1A shows that the growth rate is affected for maybe 30 minutes after exposure to 10 mM hydrogen peroxide, but then largely recovers subsequently. Why is that not reflected in the recovery of ATP levels at later times after exposure to 10 mM hydrogen peroxide (Fig. 3)?

3. With reference to the rirA deletion mutant, it would be helpful and important to provide some information regarding its phenotype. Does it grow like the wild type, and to the same density? Perhaps a growth curve relative to wildtype could be presented in a figure along with the Table 1 details.

4. The results with the mutant are difficult to grasp as written (lines 401- 459), and I am afraid the impact would be lost on the reader. The authors should make an effort to integrate these results with those of the wild type in a clearer way. Fig. 2B very nicely displays changes in specific proteins over time in response to oxidative stress (although I had trouble reading the font clearly). I would suggest including in Fig. 2B corresponding changes in the mutant – this will make clear to the reader at a glance which of the stress-responsive proteins are dependent on RirA. At the very least, this should be done with regard to iron metabolism (lines 443-449).

Additional points:

i. Line 97: “Rodeobacter”- a typo?

ii. Lines 144-145: The sentence was not clear to me: do the two samples indicate two out of three biological replicates?

iii. Lines 167-169: the sentence does not read well?

iv. Lines 169-170: it would be useful to state here the type of oxidative stress induced by each compound (peroxide, thiol etc.) for the benefit of the reader.

v. Line 199. Please state the statistical test used.

vi. Line 210: is it 63 proteins, not 73 that are changed with two or three of the oxidants?

vii. Line 231, one “to” needs to be deleted.

viii. Fig. 2B. Text is not clear. Please explain color code in the legend.

ix. Lines 263-264. It may be nice to have the data for these regulators presented in a figure. Perhaps it is included in the Iron metabolism section of Fig. 2B, but it is not legible.

x. Lines 270-272. Since a reference to OxyR is made, please comment on presence or expression of an OxyR homolog in D. shibae.

Reviewer #2: This paper examines the oxidative stress response of Dinoroseobacter shibae by looking at protein expression upon exposure to hydrogen peroxide, paraquat and diamide. Identified proteins included those involved in oxidative stress defense as well as a number of iron-regulated proteins, including RirA. The paper then makes an effort to identify RirA-regulated proteins involved in oxidative stress defense.

General issues - use of language. Aside from comma use errors, some run-on sentences, and awkward phrasing, there are a number of areas where the word choice is incorrect. The document should be close reviewed for grammar and clarity.

A few selected examples

line 110 - “exposition to stress” should be “exposure to”

line 128 -“Peptides from each gel piece were solved in..” should be “purified using”

line 142 – “labelfree” should be “label-free”

line 144 - “A protein was considerably” should be “considered”

Line 157 - “exposition” again

Line 167-168 - “ sole carbon source in the dark, different” should be “…in the dark. Different…”

Line 180-181 - “in the presence of diamide of unstressed cells” – needs to be clarified

Line 214 – “different ways in D. shibae, this was…” should be “…in D. shibae. This was..”

Line 217 – ferritine should be ferritin

Line 324-325 “ and gluconeogenesis has been shown to be a subject” should be “and gluconeogenesis have been shown to be subjects”

Figure 2b – Iron metabolism. The first item is “Soul heme-binding protein”. I suspect you meant “sole”

Experimental – The protein expression data is all represented reasonably well in the figures and accompanying text, as is the analysis of ATP levels. I am not confident at all in the data presented for the RirA mutant in response to peroxide stress (Tab. 1 line 430-435). To begin with, a table is not the best way to present these data, particularly when you are looking at ranges of time (0-30, 30-120 min). A line graph would be best here. Additionally, there are some points in this figure that call it’s accuracy into question. Notably, the wild-type and complemented mutant showed negative optical density at the 0-30 minute time point. Also, none of the control strains showed any significant growth from the 0-120 minutes, but all samples showed increased growth in the same time period when exposed to peroxide. Increased survival might be a likely outcome for the rirA mutant, based on what you are proposing, but it is highly unlikely that oxidative stress is promoting growth for all strains involved.

For Figure 4, the way the data are presented does not make it clear what you are trying to show. Rather than just accession number examples of several representatives from each cluster, I would suggest a more in-depth analysis of what your representative proteins are predicted to do, how they are differentially regulated in the rirA mutant and why that change in regulation is associated with the observed rirA mutant phenotype. This figure just feels like a collection of information that has not been sufficiently analyzed for meaning.

6. PLOS authors have the option to publish the peer review history of their article (what does this mean?). If published, this will include your full peer review and any attached files.

Reviewer #1: No

Reviewer #2: No

---

## [Author Response · Author response to Decision Letter 0]

12 Feb 2021

Our specific response to the reviewer comments are submitted as an extra file.

---

## [Decision Letter · Decision Letter 1]

8 Mar 2021

Adaptation of *Dinoroseobacter shibae* to oxidative stress and the specific role of RirA

PONE-D-20-31011R1

Dear Dr. Engelmann,

We’re pleased to inform you that your manuscript has been judged scientifically suitable for publication and will be formally accepted for publication once it meets all outstanding technical requirements. Reviewer 2 has offered some good suggestions for improving the presentation, but at this point I will leave it up to  your whether or not you decide to revise the final version of  the paper accordingly. 

Thanks again for submitting your work to PLOS ONE!

Sincerely,

Marty Roop

Academic Editor

PLOS ONE

Additional Editor Comments (optional):

Reviewers' comments:

Reviewer's Responses to Questions

**Comments to the Author**

1. If the authors have adequately addressed your comments raised in a previous round of review and you feel that this manuscript is now acceptable for publication, you may indicate that here to bypass the “Comments to the Author” section, enter your conflict of interest statement in the “Confidential to Editor” section, and submit your "Accept" recommendation.

Reviewer #1: All comments have been addressed

Reviewer #2: (No Response)

2. Is the manuscript technically sound, and do the data support the conclusions?

Reviewer #1: Yes

Reviewer #2: Yes

3. Has the statistical analysis been performed appropriately and rigorously? 

Reviewer #1: Yes

Reviewer #2: Yes

4. Have the authors made all data underlying the findings in their manuscript fully available?

Reviewer #1: Yes

Reviewer #2: Yes

5. Is the manuscript presented in an intelligible fashion and written in standard English?

Reviewer #1: Yes

Reviewer #2: Yes

6. Review Comments to the Author

Reviewer #1: (No Response)

Reviewer #2: This revision represents a much cleaner, more concise presentation of a very interesting data set. There are still a few areas where minor grammar issues need to be addressed for clarity. For example, In line 202, the word "already" is unnecessary. In line 229 "The amount of 11 additional proteins was..." could simply read "Eleven additional proteins were.." There is a similar example in line 464. In line 435, "led" should be "lead". Additionally, there are several places where commas should be used to avoid run-on sentences. Examples can be found on lines 218, 429 and 478. These are all minor issues.

It would be helpful to include a final paragraph in the conclusions which ties together the overall meaning of the findings in this paper, and what this new knowledge contributes to our knowledge of both RirA and Dinoroseobacter. There are several very interesting findings in your data set and it would really strengthen the paper to elaborate a bit on how these results contribute to the "big picture".

7. PLOS authors have the option to publish the peer review history of their article (what does this mean?). If published, this will include your full peer review and any attached files.

Reviewer #1: No

Reviewer #2: No

---

## [Editor Report · Acceptance letter]

16 Mar 2021

PONE-D-20-31011R1 

Adaptation of *Dinoroseobacter shibae* to oxidative stress and the specific role of RirA 

Dear Dr. Engelmann:

I'm pleased to inform you that your manuscript has been deemed suitable for publication in PLOS ONE. Congratulations! Your manuscript is now with our production department. 

Kind regards, 

on behalf of

Dr. Roy Martin Roop II 

Academic Editor

PLOS ONE